# ATP Synthase Members of Chloroplasts and Mitochondria in Rubber Trees (*Hevea brasiliensis*) Response to Plant Hormones

**DOI:** 10.3390/plants14040604

**Published:** 2025-02-17

**Authors:** Bingbing Guo, Songle Fan, Mingyang Liu, Hong Yang, Longjun Dai, Lifeng Wang

**Affiliations:** Key Laboratory of Biology and Genetic Resources of Rubber Tree, Ministry of Agriculture and Rural Affairs, State Key Laboratory Incubation Base for Cultivation & Physiology of Tropical Crops, Special Natural Rubber Processing Technology Innovation Center, Rubber Research Institute, Chinese Academy of Tropical Agricultural Sciences, Haikou 571101, China; guobingbing1989@catas.cn (B.G.); fansongle@gmail.com (S.F.); liumy2023@126.com (M.L.); yanghong@catas.cn (H.Y.); dailongjun@126.com (L.D.)

**Keywords:** ATP synthase, bioinformatics analysis, gene expression, rubber tree

## Abstract

ATP synthase is a key enzyme in photophosphorylation in photosynthesis and oxidative phosphorylation in respiration, which can catalyze the synthesis of ATP and supply energy to organisms. ATP synthase has been well studied in many animal species but has been poorly characterized in plants. This research identified forty ATP synthase family members in the rubber tree, and the phylogenetic relationship, gene structure, cis-elements, and expression pattern were analyzed. These results indicated that the ATP synthase of mitochondria was divided into three subgroups and the ATP synthase of chloroplast was divided into two subgroups, respectively. ATP synthase in the same subgroup shared a similar gene structure. Evolutionary relationships were consistent with the introns and exons domains, which were highly conserved patterns. A large number of cis elements related to light, phytohormones and stress resistance were present in the promoters of ATP synthase genes in rubber trees, of which the light signal accounts for the most. Transcriptome and qRT-PCR analysis showed that HbATP synthases responded to cold stress and hormone stimulation, and the response to ethylene was most significant. HbMATPR3 was strongly induced by ethylene and salicylic acid, reaching 122-fold and 17-fold, respectively. HbMATP7-1 was 41 times higher than the control after induction by jasmonic acid. These results laid a foundation for further studies on the function of ATP synthase, especially in plant hormone signaling in rubber trees.

## 1. Introduction

ATP (adenosine triphosphate) is the energy currency of the cell. ATP synthesis is mainly achieved through oxidative phosphorylation, and the terminal reaction of oxidative phosphorylation is catalyzed by ATP synthase [1,2,3]. ATP synthase is also known as F0F1-ATPase which is a multi-subunit complex that is widely distributed in the biological world, including the plasma membrane of bacteria, the inner membrane of mitochondria, and the thylakoid membrane of chloroplasts. ATP synthase has attracted significant attention in various fields ranging from bioenergetics and biophysics to chemistry, physics, and nanoscience [4,5,6,7]. Plant secondary metabolism is inevitably accompanied by the synthesis, transformation, and utilization of energy to provide necessary energy [8,9,10].

Mitochondria are commonly referred to as the “power plant” of the cell, as they are the main site where eukaryotic organisms carry out oxidative metabolism, and they are the places where carbohydrates, fats, and amino acids are ultimately oxidized to release energy [11,12]. Mitochondrial ATP synthase belongs to F0F1 ATPase in mitochondria that catalyzes ATP synthesis. It uses proton electrochemical gradients in oxidative phosphorylation through the intima, while ATP synthase is the key enzyme in the oxidative phosphorylation process [13,14,15]. Chloroplasts are important organs for photosynthesis in plant cells, and a key component is the chloroplast ATP synthase, which is located in the thylakoid membrane of the chloroplasts. Chloroplast ATP synthase is a very important enzyme system in plant cells that is capable of catalyzing the synthesis of ATP effectively [16,17,18]. Chloroplast ATP synthase belongs to the F-type ATP synthases. The ATP synthesized by chloroplasts through photosynthesis is the energy-carrying molecule. The process of coupling photochemical electron transport and phosphorylation to catalyze the reaction of ADP and Pi to form ATP in photosynthetic organisms using light energy is called photophosphorylation, which can provide energy sources for photosynthetic organisms. It is one of the most important chemical reactions on Earth. Therefore, the chloroplast ATP synthase is the key enzyme for photophosphorylation, which ultimately enables photosynthetic organisms to convert light energy into chemical energy [19,20,21,22].

Research has shown that plants need a lot of energy during growth and development and secondary metabolism, and male sterility causes ATP synthesis to be blocked, resulting in energy loss, and then it induces male sterility. The gamma subunit of chloroplast ATP synthase was two genes in Arabidopsis (*AtATPC1* and *AtATPC2*), and researchers isolated a T-DNA insertion mutant dpa1 in which growth inhibition and chloroplast morphological change phenotypes were observed. Moreover, the mutation of ATPC1 resulted in ATP synthase gamma subunit deficiency, leading to a loss of ATP synthesis and abnormal non-photochemical chlorophyll fluorescence quenching [23,24]. The activity of ATP synthase is closely related to low-temperature stress. Under low-temperature stress, ATP synthase activity decreases and ATP content decreases. Upon return to normal temperature, ATP synthase activity recovers. Rice protein kinase CTB4a enhances ATP synthase activity and ATP content by binding to atpB (ATP synthase CF1 β subunit) and thus regulates rice tolerance to low temperatures [25]. The expression of atpA was significantly induced after 96 h treatment by gray mold in tomato, and the overexpression of atpA was beneficial in resisting the gray mold infection [26]. Chloroplast ATP synthase activity was decreased by water stress, resulting in the reduction in ATP content with a lower RuBP supplement, which inhibited plant photosynthesis [27]. Sequencing of the chloroplast genome and transcriptome of cucumber revealed a maternally inherited non-synonymous single nucleotide polymorphism (SNP) in the chloroplast atpB gene, which converts threonine to arginine. The protein models showed that this change is located at the interface between the α and β subunits of ATP synthase. Polymorphism of the ATP synthase complex had been related to stress resistance in plants, which was improved by chloroplast transformation, gene editing or the creation of polymorphic β-subunit proteins [28]. There are five cotton fiber-related editing sites in the mitochondria Ghatp1 gene, and editing at positions C1292 and C1415 can significantly promote the interaction between the α and β subunits of the ATPase, enhance ATPase activity, increase energy storage, restore normal growth to yeast mutant cells, promote the increase in the number of trichomes on the surface of Arabidopsis leaves and the elongation of young roots, and significantly increase the length of cotton fiber cells by increasing ATP content [29].

The rubber tree is native to the Brazilian Amazon region, and it was called the ‘weep tree’. Natural rubber is a type of resource with high industrial and strategic value obtained from the secondary metabolites of rubber-producing plants. It is widely used in the production of the chemical industry [30]. Overall, 99% of the world’s natural rubber comes from the Brazilian rubber tree due to its high latex production, excellent quality, low cost, wide applicability, long economic life, and ease of harvesting and processing, making it the most popular variety among rubber-growing countries worldwide. As an important economic crop in tropical and subtropical regions, it is currently widely planted in tropical regions around of the world [31,32]. The synthesis of natural rubber and the regeneration of latex require energy input, so starting from the energy synthesis pathway could be a breakthrough in improving rubber biosynthesis. ATP synthase catalyzes the synthesis of the energy molecule ATP within the cell. At present, the biosynthesis process of natural rubber is very clear, but the molecular mechanism involved in energy metabolism in the synthesis process has not been studied. Based on the genome database of rubber trees, nineteen and twenty-one HbATP synthases were screened in the mitochondria and chloroplast of rubber trees, respectively. The candidate members were then analyzed by bioinformatics to provide a foundation for further research on the potential function of ATP synthase in natural rubber biosynthesis to increase the yield and quality of latex.

## 2. Results

### 2.1. ATP Synthase Family Members in Rubber Tree

According to BLAST and HMMER, these two methods, combined with Pfam and SMART, shorter sequences, and repeat sequences were deleted, and the sequence containing at least one ATP synthase domain was retained. Nineteen and twenty-one members of the ATP synthase family were identified in chloroplast and mitochondria, respectively, in the rubber tree (Table 1). Through the physical and chemical properties analysis of forty HbATP synthases, it could be seen that the amino acid length of forty ATP synthase family members ranged from 70 (HbMATPG2) to 1,057 (HbMATP1-1) aa, the molecular weight is from 6, 960.17 (HbCATPH1) to 116, 803.86 (HbMATP1-1) Da, the isoelectric point range is from 4.55 (HbCATPA3) to 9.82 (HbCATPI2), and sixteen HbATPs are unstable proteins. As the instability index surpasses 40, the value of GRAVY indicates that the majority of HbATP synthases are hydrophilic proteins with fine thermal stability.

### 2.2. Phylogenetic Analysis of ATP Synthase Gene Family in Rubber Tree

To better understand the evolutionary origin of ATP synthase and predict the potential functions, the protein sequences of forty HbATP synthases and twenty-nine AtATP synthases were clustered together to analyze the evolution of the ATP synthase in the rubber tree. The result showed that ATP synthases can be divided into three subgroups according to the evolutionary relationship (Figure 1). The ATP synthases of mitochondria in subgroup I contained the most members (eighteen), subgroup II was the second, and the minimum quantity was ten in subgroup III. There were sixteen ATP synthase members of chloroplast in subgroup I and twelve members in subgroup II.

### 2.3. Sequence Alignment and Gene Structure of ATP Synthase in Rubber Trees Members

To deeply understand the evolutionary origin and predict the function of ATP synthase, the ATP synthase domain from forty ATP synthase amino acid sequences was compared to analyze their evolution in rubber trees. The results indicated that the conserved domain in HbATP synthase is dissimilar and that there may be evolutionary differences leading to the specific differentiation (Figure 2). The gene structure analysis of HbATPs declared that the number of introns in mitochondria ranged from one to eight with HbMATP1-2 having none. ATP synthase genes in the same subgroup share a similar gene structure. Meanwhile, in the chloroplast, only *HbCATPC2* and *HbCATPD2* had upstream and downstream introns, *HbCATPC4*, *HbCATPI2*, *HbCATPH2* had two introns, *HbCATPI1*, *HbCATPD3*, *HbCATPE2* each had one intron, and the others had zero (Figure 3).

### 2.4. Cis Elements of ATP Synthase Promoter Analysis in Rubber Tree

In order to further comprehend the biological function of the ATP synthase family in rubber trees, a 2000 bp promoter sequence was extracted and twenty-six cis elements were identified in this study. The results showed that these cis elements are related to abiotic/biotic stress, light, phytohormone responsive, plant growth and development (Figure 4). All prompters of ATP synthase in the rubber tree included light, MYB, and phytohormone elements, and light elements had the largest number. Phytohormone cis elements such as ABRE, AuxRE, TGACG-motif, ERE, TCA element, and the like appeared in all members, while only a few members had the element involved in circadian. STRE was the most widely distributed defense element. There was also an as-1 element that participated in the oxidative stress response. MYB and MYC elements were present in all members except for *HbMATP8* and *HbCATPI1*, which were the two genes that did not contain the MYC element. In addition to the above response elements, a few HbATP synthase promoters also contained meristem, endosperm, and seed-specific elements.

### 2.5. Expression Analysis of HbATPs Genes

Based on the rubber tree transcript database, the expression patterns of HbATP synthases were analyzed and visualized in heat maps. Tissue expression showed that the expression level of HbATP synthase was significantly different in different tissues (Figure 5a). In this result, *HbMATPB1* and *HbMATP1-1* had high expression levels in the bark. Seven members were highly expressed in latex, corresponding to the expression level in primary and secondary laticifer, which is the synthetic tissue of natural rubber. Many HbATP synthases were expressed in latex tissues of different rubber tree cultivars, and the expression pattern was similar among cultivars. RRII105, a variety from India, showed that the number of HbATP synthases expressed was lower than other varieties. *HbMATPR3* and *HbMATP8* were only expressed in the latex of RRIM928 (Figure 5b). In the development stages of the leaves, the bronze stage and the stable stage, requiring much energy supplied, possessed the majority of ATP synthases. In the bronze stage, the energy came from the mitochondria as the leaves were not fully developed, and in the stable stage, the energy came from the chloroplast, as the leaves had become mature leaves (Figure 5c). With cold treatment of 0 h and 2 h stages, ATP synthases were concentrated and expressed in the chloroplast that leaves still photosynthesized, in cold over 8 h and 24 h, the photosynthetic system was damaged in leaves, and mitochondrial elements played a major role in energy supply; to defend against the cold condition, ATP synthases were centralized in the mitochondria (Figure 5d).

### 2.6. ATP Synthases Were Induced by Different Phytohormones

A great number of phytohormone cis elements have been predicted in the promoters of the HbATP synthase family; different phytohormones were treated in rubber trees to analyze the expression level by qRT-PCR. The results revealed that ATP synthase can be stimulated by ETH, JA, and SA at different processing times. With ETH treatment, mitochondria ATP synthases were all up-regulated, and peak values appeared in 10 h (Figure 6a). While in chloroplast, only a few ATP synthases such as *HbCATPH1*, *HbCATPH2*, *HbCATPC3*, *HbCATPC4*, *HbCATPD1*, and *HbCATPD3* were down-regulated by ETH (Figure 7a). In JA stimulation, the gene expression of ATP synthases in the rubber tree was increased, and the summit value arose in 10 h except *HbMATPR1*, *HbMATPR2*, *HbMATPR3*, and *HbMATPR4*; these four genes were reduced by JA (Figure 6b and Figure 7b). Under SA processing, apart from *HbCATPI1*, *HbCATPI2*, *HbCATPC1*, *HbCATPE2*, *HbCATPF*, *HbCATPB*, *HbCATPH1*, *HbCATPD2*, *HbCATPE1*, *HbCATPE3*, *HbMATP7-1*, *HbMATP8*, *HbMATPB1*, *HbMATPB2*, and *HbMATPB3*, other HbATP synthases were increased stimulus, and the peak occurred at 6 h (Figure 6c and Figure 7c).

## 3. Discussion

ATP (adenosine triphosphate) can trigger calcium transduction signals, which play a role in regulating the organism’s resistance to harsh environments, and a lack of ATP can lead to conditions such as organism failure. The two most common methods of ATP energy conversion are oxidative phosphorylation and photophosphorylation [33,34]. ATP synthase is mainly involved in antioxidant reactions and photophosphorylation reactions. ATP synthase deficiency in the chloroplast prevented photosynthetic autotrophy and eventually led to plant death [18,35,36]. Reports showed that many tumor cells contain ATP synthase, but because the content on the cell surface was not the same, studying ATP synthase could achieve the effect of tumor treatment [35]. ATP synthase was concentrated on lipid rafts: a microdomain on the cell surface. During lipid metabolism, ATP synthase inhibitors and anti-α and β subunit antibodies inhibited the formation of lipid droplets, providing the basis for the development of anti-obesity drugs [33,37]. However, there are few studies on ATP synthase and genes in plants, which is our deficiency. In summary, we concluded that ATP synthase is very important for animals, plants, and bacteria to regulate body health and resist adverse factors. As an energy currency, ATP is a widely used energy carrier in plant cells that is involved in various life activities during plant growth. In addition, ATP is not only an important signal that triggers calcium conduction but also activates the signaling regulatory system of the MAPK cascade, which plays a regulatory role in the resistance of plants to different harsh environments [38]. Some cellular responses to ATP in plants are similar to those in mammals, such as increases in cytoplasmic calcium, reactive oxygen species, nitric oxide production, and the role of ecto-apyrases in regulating extracellular ATP homeostasis [39]. In wild-type Arabidopsis seedlings expressing the calcium reporter protein aequorin, exogenously applied ATP triggered cytoplasmic calcium influx [40]. In the mutant of *atdorn1-1*, the addition of ATP did not trigger the phosphorylation of mitogen-activated protein kinase 3 (MPK3) and MPK6 in wild-type plants [38]. In plant cells, ATP is catalyzed by ATP synthase, which is a co-factor and important kinase [41].

Rubber trees have been important industrial and economic crops in the world. The analysis of energy synthesis is of great significance for realizing the rapid growth and high yield of rubber trees for the sustainable development of the rubber industry and meeting the market demand. In this research, twenty-one and nineteen ATP synthases were identified in the mitochondria and chloroplasts of rubber trees, respectively, and twenty and nine ATP synthases were identified in the mitochondria and chloroplasts of Arabidopsis. This result manifested that ATP synthase was conserved in mitochondria, while in chloroplasts, there was an evolution and duplication of ATP synthase, illustrating that ATP synthase in different tissues may exist in functional differentiation. These forty ATP synthases in the same tissue could be divided into four subgroups. The gene structure is crucial in exploring the evolutionary relationships of genes [42]. Members of the same subgroup had a similar gene structure and protein structure. The number of exons affects gene evolution, and the number of introns determines the rate of gene expression. Structurally similar genes often have similar functions, so structural analysis can be used to predict gene function. The results showed that most of the HbATP synthase genes in the same branch have similar exon and intron numbers, suggesting that these members have identical functions. ATP synthases in mitochondria and chloroplasts share significantly different gene structures, and studies have shown that genes that respond quickly to stimuli are more likely to have genes without introns, which can delay regulatory translation and rapidly regulate the entire growth and development process [43]. These findings indicate that ATP synthase in chloroplasts could respond more rapidly than mitochondria due to evolutionary selection for species to adapt to changes in the external environment over a long period of time. Cis elements bind to corresponding genes and play a key role in signaling and regulating the initiation of gene transcription by acting as molecular switches [44]. By identifying the cis elements in the upstream of ATPase genes in the rubber trees, the mechanisms regulating the expression of each ATPase could be better comprehended. The results showed that most of the promoter sequences of HbATP synthase contained hormone response elements, light response elements, abiotic stress elements, regulation of plant growth cycle, secondary metabolism, and other elements, suggesting that ATPase in rubber trees plays an important role in regulating the growth and development and adapting to the environment. Under various stresses, these cis elements might regulate the expression of the HbATP genes by binding to specific transcription factors, thus enabling maize to resist various abiotic and biotic stresses. In addition, HbATP synthases were more prominent in environmental adaptation and stress response, and they have more complex mechanisms in response to plant hormones and abiotic stresses.

The expression pattern of ATP synthase in rubber trees was spatially and temporally specific. In this study, we analyzed the expression of ATP synthase in rubber trees in different tissues, varieties, developmental stages, and cold stress, and we concluded that the expression patterns of HbATP synthase genes were different in different varieties, different tissues, different stress times and different stages, and they might play different roles in plant growth and development and in coping with environmental stress. A lot of HbATP synthases were expressed in secondary laticifers, indicating that HbATP synthases participated in natural rubber biosynthesis. RRIM600 is the ancestor of CATAS7-33-97 and CATAS7-33-97 is the ancestor of CATAS8-79. From the expression pattern of HbATP synthase in latex, we can see that the ATP synthase family is in the process of continuous evolution with the development of progeny. In the bronze stage and the stable stage of leaf development, the expression pattern of HbATP synthase was opposite, demonstrating that the ATP synthase of rubber trees has different functions at different developmental stages. With cold treatment, at the stage of 24 h, impaired leaf function prevented photosynthesis, so ATP synthase is only expressed in mitochondria to resultant energy. Plant hormones are involved in the process of latex production and the flow in rubber trees as regulation agents [45,46,47]. Exogenous ethylene can inhibit the competitive substances of ethylene synthesis in plants, promote the synthesis of endogenous ethylene in plants, and play a significant regulatory role in plant growth under development and adversity stress. Ethylene could increase the yield of rubber trees by prolonging the time of the latex flow period, and it was widely used in production as a stimulant to increase production but had no effect on the secondary laticifer differentiation in rubber trees [48,49,50]. JAs can induce stomatal opening, inhibit Rubisco synthesis, and affect the absorption of nitrogen and phosphorus and the transport of organic matter such as glucose [51]. JA often works in conjunction with other plant hormones to regulate each other, allowing plants to adapt to changing environmental conditions [52]. Jasmonic acid could promote the secondary laticifer differentiation in the rubber trees and regulate the natural rubber biosynthesis positively [53,54]. Salicylic acid was involved in the regulation of many physiological and biochemical processes in plants, including the improvement of plant resistance to biotic or abiotic stress and the regulation of the biosynthesis of plant secondary metabolites [55]. Natural rubber belongs to terpenoids with salicylic acid mainly affecting the accumulation of target products by regulating the activity of key enzymes, antioxidant enzymes, coding gene expression, transcription factors, and promoter activity of terpenoid biosynthesis [56]. In this research, the expression level of *HbMATPR3* and *HbCATPA3* was significantly increased by ETH. *HbMATP7-1* and *HbCATPC2* upregulation and *HbMATPR1* and *HbMATPR2* downregulation was the most significant in JA treatment. Under SA treatment, *HbMATPR3* and *HbCATPC2* were the most upregulated. We speculated that these genes play an important role in the hormone-stimulated latex production and latex flow process, and their functions could be further analyzed in the future.

## 4. Materials and Methods

### 4.1. Identification of ATP Synthase Members in Rubber Trees

The genome database and transcriptome data were obtained from the Rubber Research Institute. Based on the amino acid sequences of ATPase in Arabidopsis, BLASTP was conducted in the rubber tree genome database to capture homologous sequences to ensure the candidate members of HbATP synthase. HMMER (https://www.ebi.ac.uk/Tools/hmmer/, accessed on 10 March 2024) was used to verify whether the protein domain of these candidate members belonged to the ATP gene family and all members of the ATP synthase family in the rubber tree were obtained. According to the sequences we screened, ExPASy (https://www.expasy.org/, accessed on 3 August 2024) was used to predict the physicochemical properties of the HbATP synthases, including the amino acid number, isoelectric point, protein relative molecular weight, instability index, aliphatic index, and grand average of hydropathy. WoLF PSORT (https://www.genscript.com/wolf-psort.html, accessed on 15 August 2024) was used to predict the subcellular location of HbATP synthases.

### 4.2. Sequence Alignment and Phylogenetic Tree Reconstruction of ATP Synthase in Rubber Tree

The phylogenetic tree is used in biology to represent the evolutionary relationships between species. To investigate the relationship of ATPs in genetic evolution, Clustal W was used to align MEGA 11 software with the neighbor-joining method and p-distance. The bootstrap set to 1000 to reconstruct a phylogenetic tree for amino acid sequences of ATP in rubber tree and Arabidopsis. The multiple sequence alignment was beautified by Jalview v2.11 software.

### 4.3. Gene Structure and Promoter Analysis of HbATP Synthase

The main purpose of gene structure analysis is to identify gene and non-coding DNA regions and to study the relationships with associated functions between gene family members. The exon–intron gene structure of rubber tree ATP synthase genes was analyzed through the Gene Structure Display Server GSDS 2.0 (https://gsds.gao-lab.org). A total of 2000 bp was extracted in the upstream of the genomic sequences of HbATP, and PlantCARE (https://bioinformatics.psb.ugent.be/webtools/plantcare/html/) was used to analyze the cis element with visual display in TBtools v2.149 and statistics in GraphPad Prism v9.5 software.

### 4.4. Gene Expression Pattern of ATPase in Rubber Trees

The RNA-seq data of CATAS7-33-97 were obtained in HeveaDB (http://hevea.catas.cn/home/index) to screen the FPKM (Fragments Per Kilobase of transcript per Million mapped reads) values of HbATP in different tissues, varieties, developmental stages and treatments. The data obtained after log2 normalization were used to generate gene expression heat maps and standardized through the scale method (zero to one) using the module Heatmap Illustrator in TBtools.

### 4.5. Plant Materials and Treatment

The experimental materials used in the hormone stimulation of expression were all from the rubber tree tissue culture seedings ‘CATAS7-33-97’ that were planted in the experimental field of the Chinese Academy of Tropical Agricultural Sciences (Danzhou City, Hainan Province, China). First, 1.5% ethephon (ETH), 200 μmol/L jasmonic acid (JA), and 200 μmol/L salicylic acid (SA) were sprayed until the whole plants were moist with 0.05% alcohol for control, and tissue culture seedlings were treated for 0, 0.5, 2, 6, 10, and 24 h to collect the leaves and freeze with liquid nitrogen. All samples were stored at −80 °C. Each sample in this research had three biological replicates.

### 4.6. qRT-PCR Analysis of HbATPs in Different Plant Hormones Induction

All plant material samples were used for RNA extraction by FastPure Complex Tissue/Cell Total RNA Isolation Kit (Vazyme, China) according to the instruction procedures with 75% ethanol cleaned in the experimental area and sterilized spatulas, spoons, etc., while the liquid nitrogen was then divided into insulated pots for later use. The total RNA concentration and purity were determined using a NanoDrop 2000 analyzer (Thermo Fisher, Waltham, MA, USA), and the RNA was stored at −80 °C in the refrigerator for later use. The RNA reverse transcription was carried out according to the experimental steps of the Thermo Scientific RevertAid RT Kit (Thermo Fisher, Waltham, MA, USA), and the quality of cDNA was detected using HbActin as an internal reference gene. The integrity of the RNA and the concentration and purity of the cDNA were detected by 1.5% agarose gel electrophoresis. Primers used in qRT-PCR were designed on the NCBI online website (https://www.ncbi.nlm.nih.gov/tools/primer-blast/) and shown in Appendix A. qRT-PCR was conducted in CFX Connect 96 (Biolab, Germantown, MD, USA) with 0.4 μL forward primer, 0.4 μL reverse primer, 1 μL cDNA (1 μg), 8.2 μL ddH_2_O, and 10 μL ChamQ SYBR Color qPCR Master Mix (Vazyme, Nanjing, China) by the following steps: initial denaturation at 98 °C for 2 min, denaturation at 98 °C for 15 s, annealing at 58 °C for 15 s, and extended at 72 °C for 1 min, repeated for 35 cycles.

### 4.7. Data Processing and Analysis

The expression level of *HbATP* synthase was calculated by 2^-∆∆Ct^. Each experiment consisted of three biological replicates. Graphic Prism software was used for mapping.

## 5. Conclusions

The synthesis of ATP is one of the most important chemical reactions in biological organisms. ATP synthase catalyzes the synthesis of the energy source ATP in cells. In this study, a total of forty ATP synthase gene family members were identified based on the rubber tree genome database. The phylogenetic analysis of the twenty-one ATP synthase genes in mitochondria was divided into three subgroups, while the nineteen ATP synthase genes in chloroplast were divided into two subgroups. The evolutionary and phylogenetic relationship among these groups is backed by a significant level of conservation in intron and extra-domain patterns. Transcriptome data and qRT-PCR results showed that the expression patterns of this gene family members were different under different tissues, different hormones, and stress to support more energy in natural rubber biosynthesis. Our categorization and evolutionary examination, along with the identification of motif characteristics and intron arrangements within the rubber tree ATP synthase family, can establish a strong basis for future research on ATP synthase control functions in crucial growth and developmental processes. On the other hand, these results provide a reference for further research into the function of the ATP synthase gene and breeding by gene editing.

## Figures and Tables

**Figure 1 plants-14-00604-f001:**
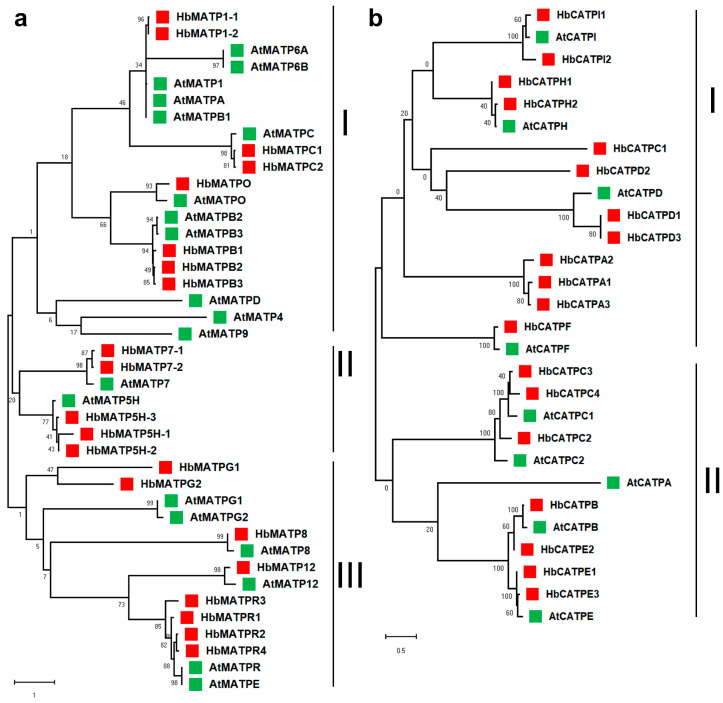
The phylogenetic tree of ATP synthase in Arabidopsis and rubber tree. Phylogenetic relationship among 69 proteins in these two species. The neighbor joining tree was generated by MRGA 11 (1000 replicates), and each branch’s bootstrap value is displayed. (**a**) ATP synthase of mitochondria from Arabidopsis and rubber tree. The red triangle represents the gene in the rubber tree and the green triangle represents the gene in Arabidopsis. (**b**) ATP synthase of chloroplast from Arabidopsis and rubber tree. The scale bars indicate 1 and 0.5 amino acid substitutions per site, respectively.

**Figure 2 plants-14-00604-f002:**
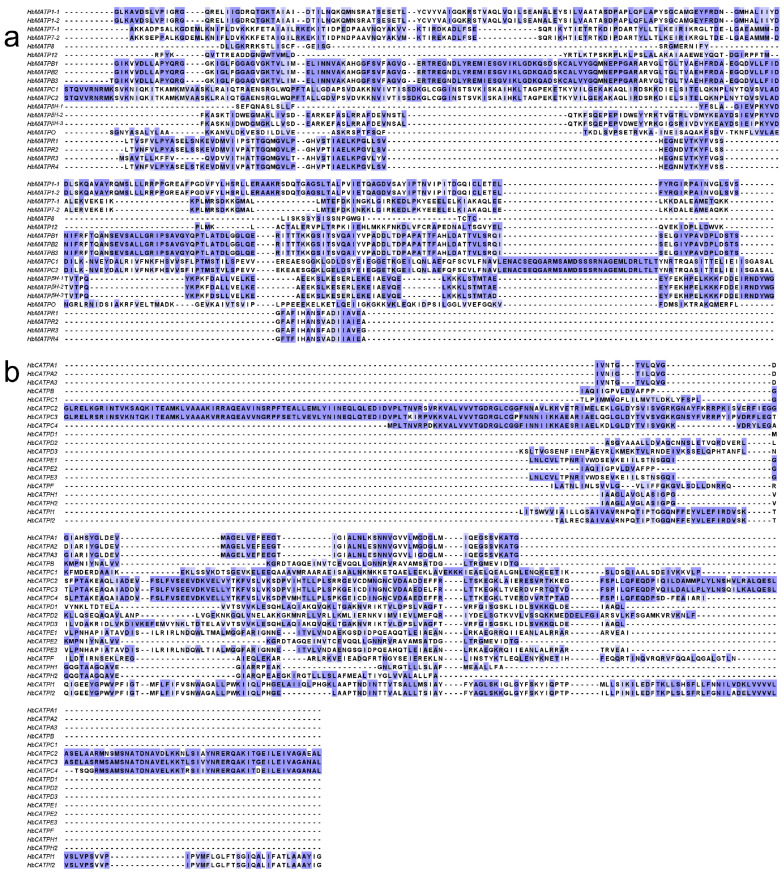
Multiple alignment of the ATP synthase domains of rubber tree. Conserved domains of the ATP synthase amino acid sequence are marked in colored percentage with the dark color indicating high similarity and light color indicating low similarity in aligned sequences.

**Figure 3 plants-14-00604-f003:**
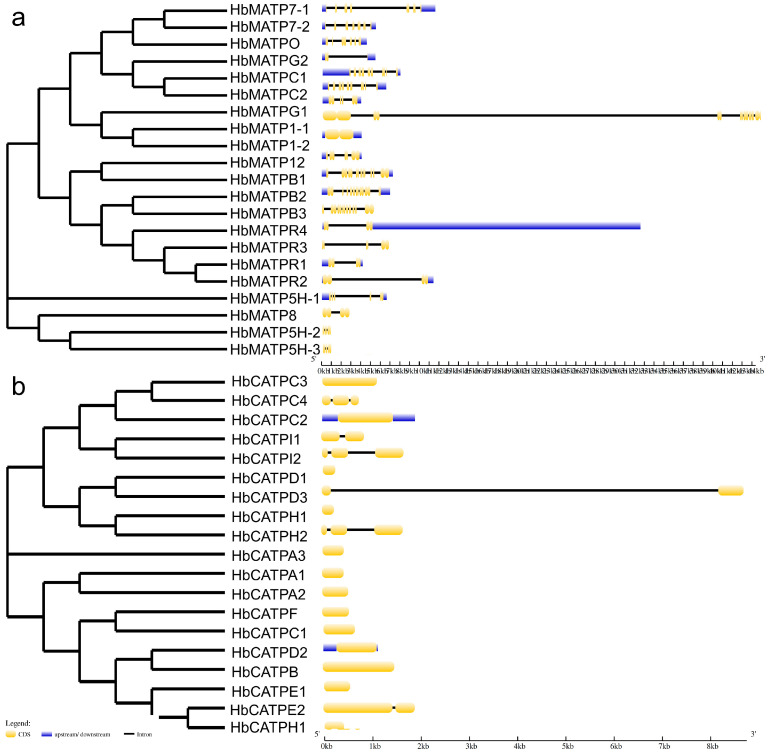
Gene structure and conserved motifs in forty ATP synthase members from rubber tree. The phylogenetic tree of ATP synthase was contrasted by MEGA 11 using the NL method. The gene structure of the ATP synthase genes was analyzed by GSDS. Yellow strips represent CDS, black lines delegate intron and blue strips state up/downstream. (**a**). Gene structure of ATP synthase from mitochondria of rubber tree. (**b**) Gene structure of ATP synthase from chloroplast of rubber tree.

**Figure 4 plants-14-00604-f004:**
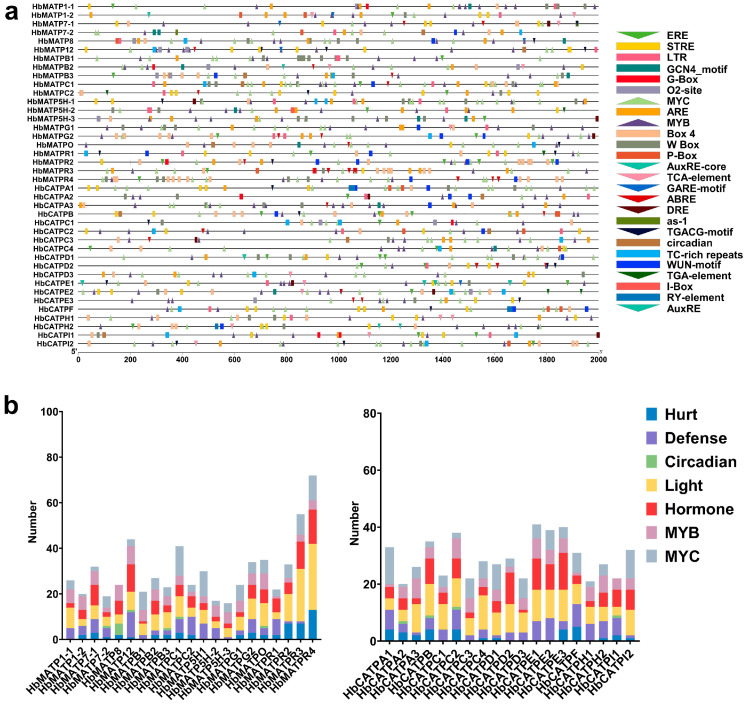
Cis-element analysis of ATP synthase in rubber tree. (**a**). Cis elements in *HbATP* synthase promoters. Left: Cis elements distributed in the promoter sequences of *HbATP* synthase. Right: Boxes in different colors represent different elements responding to different stresses. (**b**). The number of different kinds of cis elements in the promoter of ATP synthase in rubber trees.

**Figure 5 plants-14-00604-f005:**
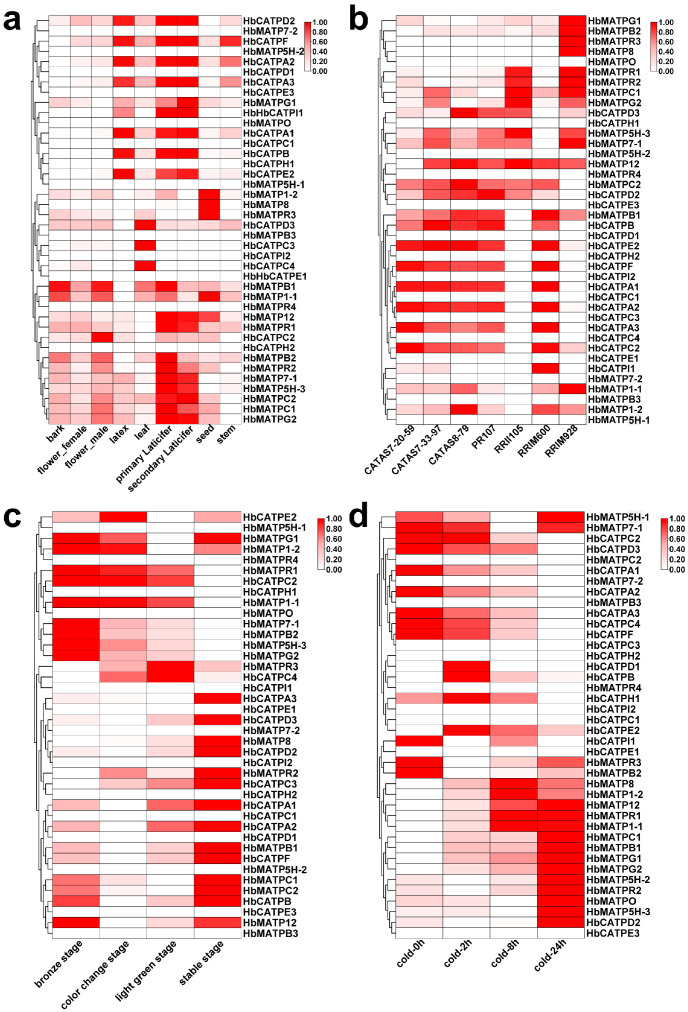
Expression patterns of ATP synthase genes in different tissues, varieties, developmental stages, and cold stress of rubber trees. *HbActin* was used as a reference gene. (**a**). The heat map of ATP synthase in different tissues of rubber trees. (**b**). The heat map of ATP synthase in different cultivates of rubber trees. (**c**). The heat map of ATP synthase in different development stages in the leaves of rubber trees. (**d**) The heat map of ATP synthase in cold condition. The depth of color shown in the heat map represents the level of expression and is standardized through the scale method (zero to one).

**Figure 6 plants-14-00604-f006:**
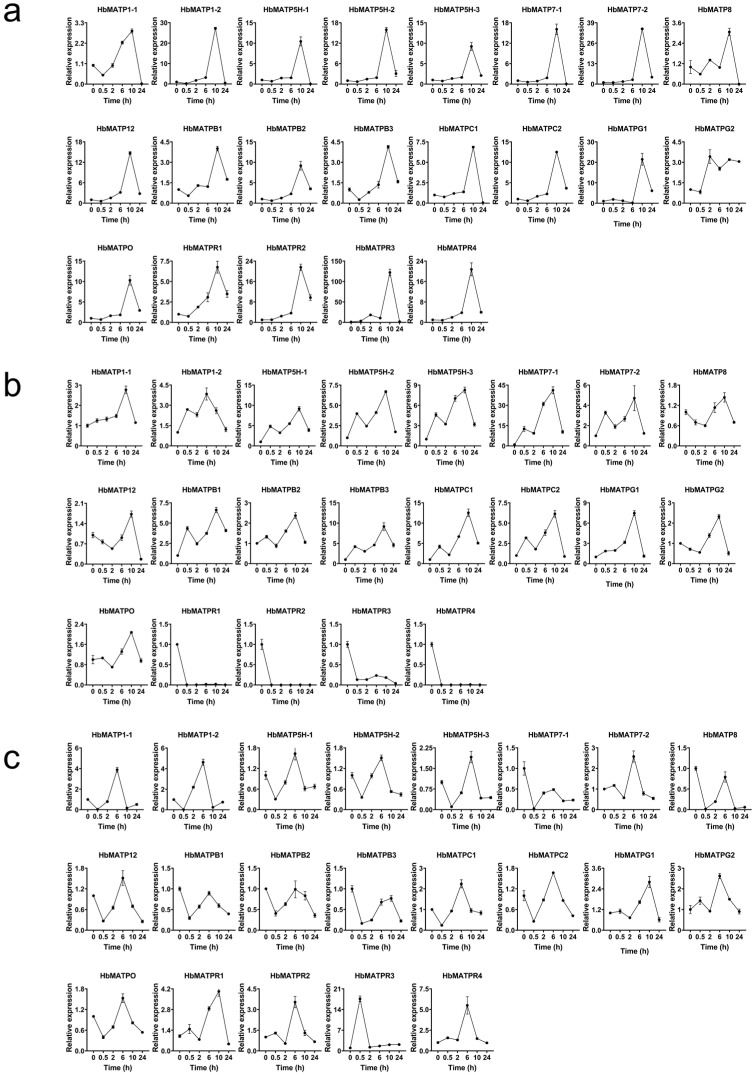
ATP synthase of mitochondria in rubber tree response to different hormones (ETH, JA, SA). (**a**). ATP synthase of mitochondria with ETH treatment. (**b**). ATP synthase of mitochondria with JA treatment. (**c**). ATP synthase of mitochondria with SA treatment. ETH, ethylene; JA, jasmonic acid; SA, salicylic acid. Error bars indicate SD to represent the standard deviation of three independent experiments. *HbActin* was the reference gene.

**Figure 7 plants-14-00604-f007:**
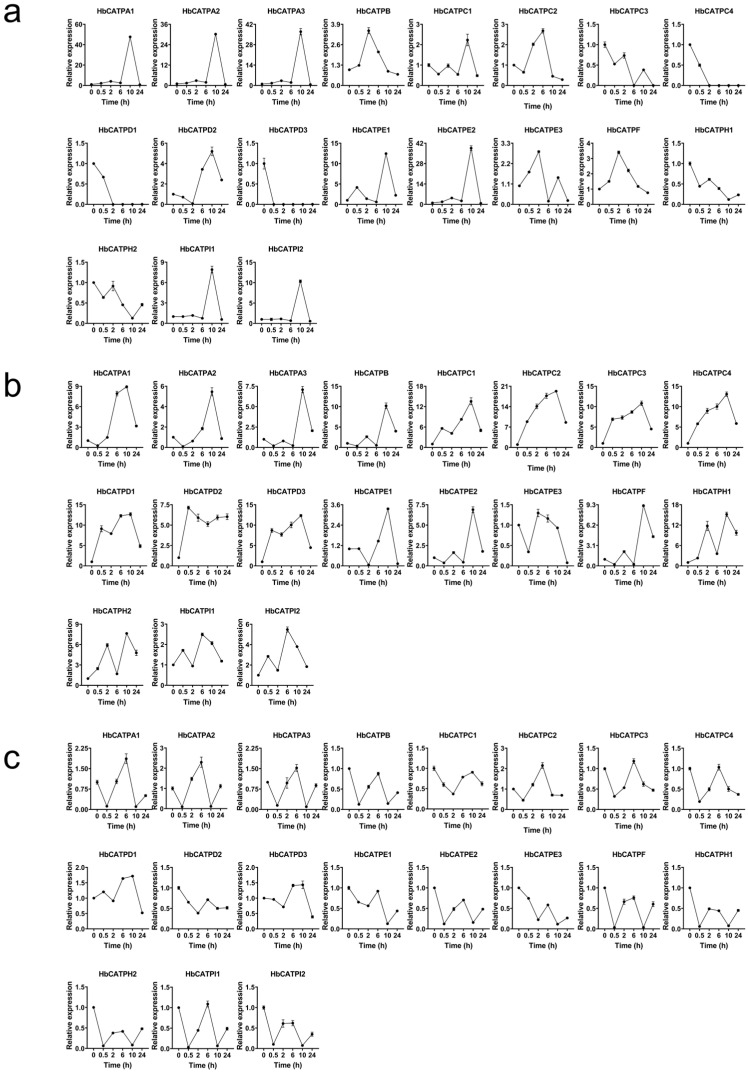
ATP synthase of chloroplast in rubber tree response to different hormones (ETH, JA, SA). (**a**). ATP synthase of chloroplast with ETH treatment. (**b**). ATP synthase of chloroplast with JA treatment. (**c**). ATP synthase of chloroplast with SA treatment. ETH, ethylene; JA, jasmonic acid; SA, salicylic acid. Error bars indicate SD to represent the standard deviation of three independent experiments. *HbActin* was the reference gene.

**Table 1 plants-14-00604-t001:** Physical and chemical properties of ATP synthase from chloroplast and mitochondrion in rubber tree.

Gene Name	CDS (bp)	Amino Acid (aa)	Molecular Weight	pI	Instability Index	Aliphatic Index	GRAVY	Locate
*HbCATPA1*	447	149	16,095.39	4.75	40.42/un	110.54	0.014	Chloroplast
*HbCATPA2*	513	171	18,678.57	4.73	45.44/un	120.23	0.201	Chloroplast
*HbCATPA3*	441	147	16,056.32	4.55	21.28	117.35	0.208	Chloroplast
*HbCATPB*	1479	493	53,120.94	5.16	37.59	98.68	−0.005	Chloroplast
*HbCATPC1*	651	217	23,745.87	5.8	40.82/un	104.88	−0.037	Chloroplast
*HbCATPC2*	1140	380	42,565.01	7.58	37.54	95.45	−0.167	Chloroplast
*HbCATPC3*	1128	376	41,303.39	6.39	45.26/un	101.6	−0.093	Chloroplast
*HbCATPC4*	687	229	25,168.82	4.97	29.16	98.34	−0.127	Chloroplast
*HbCATPD1*	255	85	9234.88	9.4	16.36	127.18	0.256	Chloroplast
*HbCATPD2*	834	278	31,274.51	9.61	51.96/un	96.37	−0.117	Chloroplast
*HbCATPD3*	624	208	22,641.02	6.16	30.56	110.58	0.041	Chloroplast
*HbCATPE1*	399	133	14,608.79	5.41	40.85/un	116.69	−0.032	Chloroplast
*HbCATPE2*	1833	611	66,044.72	5.19	38.31	101.83	−0.017	Chloroplast
*HbCATPE3*	399	133	14,529.67	5.67	34.21	118.87	0.002	Chloroplast
*HbCATPF*	552	184	20,936.82	7.87	30.21	102.28	−0.341	Chloroplast
*HbCATPH1*	213	71	6960.17	8.5	18.14	118.45	0.786	Chloroplast
*HbCATPH2*	300	100	10,504.47	8.98	21.61	128.9	0.663	Chloroplast
*HbCATPI1*	777	259	28,444.63	6.36	18.89	125.29	0.724	Chloroplast
*HbCATPI2*	1053	351	39,534.35	9.82	37.59	101.99	0.079	Chloroplast
*HbMATP1-1*	3171	1057	116,803.86	9.46	43.44/un	95.77	−0.048	Mitochondrion
*HbMATP12*	996	332	36,494.87	6.25	43.44/un	91.39	−0.121	Mitochondrion
*HbMATP1-2*	1527	509	55,254.41	5.84	37.2	100.26	−0.059	Mitochondrion
*HbMATP7-1*	726	242	28,032.83	9.14	34.81	92.4	−0.587	Mitochondrion
*HbMATP7-2*	720	240	27,857.65	9.25	35.47	91.92	−0.613	Mitochondrion
*HbMATP8*	987	329	37,439.28	9.36	48.79/un	82.01	0.065	Mitochondrion
*HbMATPB1*	1662	554	59,440.88	6.11	41.52/un	95.74	−0.09	Mitochondrion
*HbMATPB2*	1518	506	54,280.89	5.42	30.29	95.57	−0.088	Mitochondrion
*HbMATPB3*	1686	562	60,116.53	6.12	40.41/un	93.54	−0.122	Mitochondrion
*HbMATPC1*	969	323	35,223.52	8.99	51.91/un	97	−0.158	Mitochondrion
*HbMATPC2*	969	323	35,228.55	8.83	50.31/un	96.35	−0.147	Mitochondrion
*HbMATP5H-1*	363	121	14,160.23	5.16	30.1	82.23	−0.486	Mitochondrion
*HbMATP5H-2*	504	168	19,815.47	5.13	33.96	69.64	−0.848	Mitochondrion
*HbMATP5H-3*	504	168	19,696.25	5.11	33.41	71.37	−0.861	Mitochondrion
*HbMATPG1*	738	246	28,145.22	9.25	37.84	84.88	−0.261	Mitochondrion
*HbMATPG2*	210	70	7954.06	8.66	41.98/un	65.57	−0.499	Mitochondrion
*HbMATPO*	747	249	27,274.67	9.56	49.06/un	97.11	−0.148	Mitochondrion
*HbMATPR1*	600	200	21,355.57	5.79	34.35	93.65	0.074	Mitochondrion
*HbMATPR2*	843	281	30,542.57	7.85	43.42/un	106.58	0.202	Mitochondrion
*HbMATPR3*	681	227	24,832.65	5.72	19.31	91.01	0.126	Mitochondrion
*HbMATPR4*	609	203	21,777.98	5.78	33.33	84.98	−0.007	Mitochondrion

## Data Availability

Data are contained within the article and Appendix A.

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
