# Peer review of "ATP Synthase Members of Chloroplasts and Mitochondria in Rubber Trees (Hevea brasiliensis) Response to Plant Hormones"

_plants, 2025, doi:10.3390/plants14040604_

Round 1
Reviewer 1 Report
Comments and Suggestions for Authors
I checked yuor manuscript and described comments below.
In rubber trees, ATP synthase is an important enzyme that produces ATP, the energy source used to produce isopentenyl diphosphate (IPP), the raw material for rubber.
This paper performs very good analysis by performing RNA-seq expression analysis from existing databases and looking at the relationship with different hormones such as ETH, JA, and SA using qRT-PCR.
I think it would be better to consider the following points.
1. There are many kinds of rubber trees. The genetic code for this plant is Hb, so I think the scientific name is probably "Hevea benthamiana", but I think it would be better to mention this in the title or text.
2. About Figures 1 and 3, I use MEGA X to draw the phylogenetic tree. The latest version is MEGA 11. If possible, I recommend using the latest software for analysis.
3. About Figure 3, the bootstrap value is not listed. I think it would be better to fix that.
4. I think it would be better to align the vertical scales of the graphs in Figures 6 and 7 to the same scale.
5. There is no Table S1. I think it is important because it is the sequence of the qRT-PCR primer.
I don't think this paper has major problems and grammatical problems
Author Response
Response to the Review Comments
Dear Editors and Reviewer:
Thank you for your letter and the reviewers’ careful reading, helpful comments, and constructive suggestions concerning our manuscript entitled ‘ATP synthase members of chloroplasts and mitochondria in rubber trees (Hevea brasiliensis) response to plant hormones.’ (ID: plants-3429487), which has significantly improved the presentation of our manuscript. We have carefully considered all reviewer comments and revised our manuscript accordingly. The manuscript has also been double-checked. In the following section, we summarize our responses to each comment from the reviewer. We believe that our responses have well addressed all the concerns of the reviewer. We hope our revised manuscript can be accepted for publication.
Reviewer 1:
I checked your manuscript and described comments below.In rubber trees, ATP synthase is an important enzyme that produces ATP, the energy source used to produce isopentenyl diphosphate (IPP), the raw material for rubber.This paper performs very good analysis by performing RNA-seq expression analysis from existing databases and looking at the relationship with different hormones such as ETH, JA, and SA using qRT-PCR.I think it would be better to consider the following points.1. There are many kinds of rubber trees. The genetic code for this plant is Hb, so I think the scientific name is probably "Hevea benthamiana", but I think it would be better to mention this in the title or text.2. About Figures 1 and 3, I use MEGA X to draw the phylogenetic tree. The latest version is MEGA 11. If possible, I recommend using the latest software for analysis.3. About Figure 3, the bootstrap value is not listed. I think it would be better to fix that.4. I think it would be better to align the vertical scales of the graphs in Figures 6 and 7 to the same scale.5. There is no Table S1. I think it is important because it is the sequence of the qRT-PCR primer.I don't think this paper has major problems and grammatical problems
Response:
We gratefully appreciate your valuable comment. Based on your comments, we revised the manuscript and responded point by point according to the resubmitted manuscript.
- There are many kinds of rubber trees. The genetic code for this plant is Hb, so I think the scientific name is probably "Hevea benthamiana", but I think it would be better to mention this in the title or text.
Reply: Thank you for your kind reminder and we have added the Latin name of the rubber tree in the title for ‘ATP synthase members of chloroplasts and mitochondria in rubber trees (Hevea brasiliensis) response to plant hormones.’
2. About Figures 1 and 3, I use MEGA X to draw the phylogenetic tree. The latest version is MEGA 11. If possible, I recommend using the latest software for analysis.
Reply: Thank you for your good suggestion to improve our manuscript. We had to redraw the phylogenetic tree of Figure 1 and 3 by MEGA 11 and the new Figures were shown in the revised manuscript.
3. About Figure 3, the bootstrap value is not listed. I think it would be better to fix that.
Reply: Thank you point out our negligence. We added the bootstrap value in Figure 3 in the revised manuscript.
4. I think it would be better to align the vertical scales of the graphs in Figures 6 and 7 to the same scale.
Reply: We are grateful for the suggestion. The vertical scales of Figure 6 and 7 were aligned to the same scale in the revised manuscript.
5. There is no Table S1. I think it is important because it is the sequence of the qRT-PCR primer.
Reply: Thank you for your good suggestion and we submitted Table S1 in the supplemental materials that were not shown in the manuscript. We could exhibit Table SI for you.
Table S1 Primers of qRT-PCR for ATP synthase in rubber trees |
|
Primer Name |
Sequence (5'-3') |
qHbCATPA1-F |
TGGTAACCATTCGAGCCGACG |
qHbCATPA1-R |
AGCAATGCCGTCGCCTACTT |
qHbCATPA2-F |
AGAGGGAAGCTCCGTAAAAGCA |
qHbCATPA2-R |
ACCTGGAGCGGGGGATTCAA |
qHbCATPA3-F |
AACCATTCGAGCCGACGAGA |
qHbCATPA3-R |
GAGCAATGCCGTCGCCTACT |
qHbCATPB-F |
TAGCCCCTTATCGCCGTGGA |
qHbCATPB-R |
GTACGTTCGCCTACTCCGCC |
qHbCATPC1-F |
CTTTGCTCCGCCCTCTCTCG |
qHbCATPC1-F |
TGAGGGCACTGCAGGAGTCA |
qHbCATPC2-R |
CTTCTCCTTGATCGCGGCGT |
qHbCATPC2-R |
AGGGCCTCTGCTCCAGCTAC |
qHbCATPC3-F |
TGGCCAGTGAGCTTGCTTCT |
qHbCATPC3-R |
TTGCCTGGCGCTCCCTATTG |
qHbCATPC4-F |
TTGAGGGTGCCTCTCTCCCT |
qHbCATPC4-R |
GCAGCATCCACACAGTTCCCA |
qHbCATPD1-F |
CTGAGCTTGCGGTCGTGACA |
qHbCATPD1-R |
AGTGAACCCAGCCACCAAGC |
qHbCATPD2-F |
GCCAATCCGCTTGTGGGTGA |
qHbCATPD2-R |
GCACACTTCGAGCGATCCCA |
qHbCATPD3-F |
CTGAGCTTGCGGTCGTGACA |
qHbCATPD3-R |
AGTGAACCCAGCCACCAAGC |
qHbCATPE1-F |
CGATGGCTCTGATGGGTGGT |
qHbCATPE1-R |
TCGTGCCCTAGCTCGTCTGA |
qHbCATPE2-F |
TAGCCCCTTATCGCCGTGGA |
qHbCATPE2-R |
GTACGTTCGCCTACTCCGCC |
qHbCATPE3-F |
CCATGCACCTATTGCCACAGC |
qHbCATPE3-R |
TGCCTTCCGCTTTCCTCAAGT |
qHbCATPF-F |
CCTTGGGTCACTGGCCATCC |
qHbCATPF-R |
TCCACTTTCCGTAAGCGGGC |
qHbCATPH1-F |
GCCGCTTCCGTTATTGCTGC |
qHbCATPH1-R |
TTTCCCTTTGCCTCGGGTCG |
qHbCATPH2-F |
GCCGCTTCCGTTATTGCTGC |
qHbCATPH2-R |
CTCCGCCTCGGGTTGTCTTG |
qHbCATPI1-F |
CGGTGTGGAAGTAGGCCAGC |
qHbCATPI1-R |
AATAAAGGGGACCCACGGGC |
qHbCATPI2-F |
TAGCGCGGGCTGCAATAAGG |
qHbCATPI2-R |
TCGAGCCGCATCTCCTTTCG |
qHbMATP1-1-F |
AGGGCGAACACTCATCGTTT |
qHbMATP1-1-R |
TCGGAATGAGAAGGAACGCC |
qHbMATP12-F |
AACACTCGCAGTCCGATGGC |
qHbMATP1-2-F |
GCCGCTAAACGATCGGACCA |
qHbMATP12-R |
AGACGACGACGCCGAATCAC |
qHbMATP1-2-R |
TTCAACTGAGCGGCAGACCC |
qHbMATP5H-1-F |
TTGCAGGCATAGAGGTCCCA |
qHbMATP5H-1-R |
AGCAGTCATGGTGCTGAGCTT |
qHbMATP5H-2-F |
TCGTCGCGCTTTCGATGAGG |
qHbMATP5H-2-R |
ACGTTCCGACTCCTTCAGGG |
qHbMATP5H-3-F |
GCCTCCCTTCGTCGTGCTTT |
qHbMATP5H-3-R |
ACACATCCACTATGCGTGAGCC |
qHbMATP7-1-F |
CAGGAGCCGGAGCGTGTTAT |
qHbMATP7-1-R |
AGTTCGCGTCTCGATGGTGT |
qHbMATP7-2-F |
TGCCAAAGCACAATTAGAGGAGC |
qHbMATP7-2-R |
TCACATCCACCGTTGCCTCG |
qHbMATP8-F |
CGGAGCAAGGACCCCAACAG |
qHbMATP8-R |
CCATTCCTCGCGAGCCACTT |
qHbMATPB1-F |
TGGGCTCTTTGGTGGTGCTG |
qHbMATPB1-R |
CGTGCACCAGGAGGCTCATT |
qHbMATPB2-F |
GTTCACCGGAAAGGGTGCGA |
qHbMATPB2-R |
TGGTCCCTCACCTCCAGAGC |
qHbMATPB3-F |
AGAGCGCACTCGTGAGGGTA |
qHbMATPB3-R |
CACGGAAGTGCTCAGCCACA |
qHbMATPC1-F |
TCACCGCCATCCGATCCTCT |
qHbMATPC1-R |
GCTGCCATAGGCCACGAGAA |
qHbMATPC2-F |
TGGCTGCTTTGAGACGCGAA |
qHbMATPC2-R |
AGGGGTCTGCTCCTCAGCAA |
qHbMATPG1-F |
GCACCCTACTTCACCGCCTC |
qHbMATPG1-R |
GCCTCTTCATCGGCGAGCTT |
qHbMATPG2-F |
CCAATGCAGCAGTGCCGTTC |
qHbMATPG2-R |
CTCGCGAGTGAGAGCTTCGG |
qHbMATPO-F |
TGGGCGTCTCAGATCGGGTC |
qHbMATPO-R |
AGCAGGGCAAAGAACGGAGC |
qHbMATPR1-F |
TCCTGGCCCGACCCATTCTT |
qHbMATPR1-R |
CGGGGCTGCATGAAGGAGAG |
qHbMATPR2-F |
CGCACCTCCGCTTCCAATCT |
qHbMATPR2-R |
GCTGGCACATCCGTCGAGAA |
qHbMATPR3-F |
GTTGAGGGTGCGCCACTTGA |
qHbMATPR3-R |
TTGCAGTTCGGCCTGTGAGC |
qHbMATPR4-F |
GCCCGGGGTCTTATCAGTGC |
qHbMATPR4-R |
TGGTCAACTGGCACAGCCTC |
To sum up, we have made corrected modifications to the revised manuscript. Please do not hesitate to contact us if there are any questions. Thanks again to the reviewer and editor for your hard work! Best wishes to you!
Authors: Bingbing Guo, Songle Fan, Mingyang Liu, Hong Yang, Longjun Dai, Lifeng Wang

Reviewer 2 Report
Comments and Suggestions for Authors
The aim of the study was to analyze the ATP-Adenosine Triphosphate groups in the rubber tree in the Brazilian Amazon region. The authors identified a total of 40 members of the ATP synthase gene family based on the rubber tree genome database. The introduction is written correctly. However, the research hypothesis is missing, which should be supplemented - line 98. The results were developed and described correctly. The appropriate research methods were used. The literature cited in the discussion is correct. The conclusions are too general. It requires more detailed development and reference to the research results and the research hypothesis.
After correction, the work can be published in the journal Plants
Author Response
Response to the Review Comments
Dear Editors and Reviewer:
Thank you for your letter and the reviewers’ careful reading, helpful comments, and constructive suggestions concerning our manuscript entitled ‘ATP synthase members of chloroplasts and mitochondria in rubber trees (Hevea brasiliensis) response to plant hormones.’ (ID: plants-3429487), which has significantly improved the presentation of our manuscript. We have carefully considered all reviewer comments and revised our manuscript accordingly. The manuscript has also been double-checked. In the following section, we summarize our responses to each comment from the reviewer. We believe that our responses have well addressed all the concerns of the reviewer. We hope our revised manuscript can be accepted for publication.
Reviewer 2:
The aim of the study was to analyze the ATP-Adenosine Triphosphate groups in the rubber tree in the Brazilian Amazon region. The authors identified a total of 40 members of the ATP synthase gene family based on the rubber tree genome database. The introduction is written correctly. However, the research hypothesis is missing, which should be supplemented - line 98. The results were developed and described correctly. The appropriate research methods were used. The literature cited in the discussion is correct. The conclusions are too general. It requires more detailed development and reference to the research results and the research hypothesis.After correction, the work can be published in the journal Plants
Reply: Thank you for your good suggestion to improve our manuscript. In the revised manuscript, we added the research hypothesis at the end of ‘Introduction’ as ‘to lay a foundation for further research on the potential function of ATP synthase in natural rubber biosynthesis to increase the yield and quality of latex’ in line 104. The conclusion was improved and more detailed by the results and hypothesis in the revised manuscript from line 398 to line 412.
To sum up, we have made corrected modifications to the revised manuscript. Please do not hesitate to contact us if there are any questions. Thanks again to the reviewer and editor for your hard work! Best wishes to you!
Authors: Bingbing Guo, Songle Fan, Mingyang Liu, Hong Yang, Longjun Dai, Lifeng Wang

Reviewer 3 Report
Comments and Suggestions for Authors
The manuscript by Guo et al. considers the evolution, functional characteristics and response to hormones of ATP synthase in chloroplast and mitochondria in the rubber trees. The manuscript seems to be interesting. However, I have some comments.
(1) Why did authors use simultaneous treatment of plants by some hormones (ethylene, jasmonic acid, and salicylic acid)?
(2) Please, describe in more details influence of ethylene, jasmonic acid, and salicylic acid on plants in Discussion. What is effect of simultaneous treatment by these hormones on plants? What are molecular mechanisms influence on changes of ATP synthase expression.
(3) Please, describe in more details results of investigation in Abstract.
(4) Please, describe in more details results of investigation in Conclusion.
(5) Please, describe in more details the role of ATP for calcium conduction, calcium signals and MAPK cascade signals.
(6) P. 11. Lines 226-228. Why did authors describe role of ATP synthase in tumor?
(7) P. 11. Lines 229-231. Why did authors describe role of ATP synthase in lipid metabolism and development of anti-obesity drugs?
Author Response
Response to the Review Comments
Dear Editors and Reviewer:
Thank you for your letter and the reviewers’ careful reading, helpful comments, and constructive suggestions concerning our manuscript entitled ‘ATP synthase members of chloroplasts and mitochondria in rubber trees (Hevea brasiliensis) response to plant hormones.’ (ID: plants-3429487), which has significantly improved the presentation of our manuscript. We have carefully considered all reviewer comments and revised our manuscript accordingly. The manuscript has also been double-checked. In the following section, we summarize our responses to each comment from the reviewer. We believe that our responses have well addressed all the concerns of the reviewer. We hope our revised manuscript can be accepted for publication.
Reviewer 3:
The manuscript by Guo et al. considers the evolution, functional characteristics and response to hormones of ATP synthase in chloroplast and mitochondria in the rubber trees. The manuscript seems to be interesting. However, I have some comments.(1) Why did authors use simultaneous treatment of plants by some hormones (ethylene, jasmonic acid, and salicylic acid)?
(2) Please, describe in more details influence of ethylene, jasmonic acid, and salicylic acid on plants in Discussion. What is effect of simultaneous treatment by these hormones on plants? What are molecular mechanisms influence on changes of ATP synthase expression.
(3) Please, describe in more details results of investigation in Abstract.
(4) Please, describe in more details results of investigation in Conclusion.
(5) Please, describe in more details the role of ATP for calcium conduction, calcium signals and MAPK cascade signals.
(6) P. 11. Lines 226-228. Why did authors describe role of ATP synthase in tumor?
(7) P. 11. Lines 229-231. Why did authors describe role of ATP synthase in lipid metabolism and development of anti-obesity drugs?
Response:
We gratefully appreciate your valuable comment. Based on your comments, we revised the manuscript and responded point by point according to the resubmitted manuscript.
(1) Why did authors use simultaneous treatment of plants by some hormones (ethylene, jasmonic acid, and salicylic acid)?Reply: We think this is an excellent suggestion. We have explained the reason in the revised manuscript in line 306-307 ‘Plant hormones are involved in the process of latex production and the flow of rubber trees as regulation agents [45-47]’.
(2) Please, describe in more details influence of ethylene, jasmonic acid, and salicylic acid on plants in Discussion. What is effect of simultaneous treatment by these hormones on plants? What are molecular mechanisms influence on changes of ATP synthase expression.
Reply: We feel great thanks for your professional review work on our article. The effect of ethylene, jasmonic acid, and salicylic acid in rubber trees is redescribed in detail in the discussion section from line 308 to line 324. Because there are few studies on ATP synthase in plants, the molecular mechanism of the effect of plant hormones on ATP expression has not been discovered, and also, that's what I'm working for.
(3) Please, describe in more details results of investigation in Abstract.
Reply: We sincerely thank you for your valuable feedback that we have used to improve the quality of our manuscript. According to your nice suggestions, we have made the details described in Abstract in the revised manuscript with a revision model.
(4) Please, describe in more details results of investigation in Conclusion.
Reply: Thank you for your decision and constructive comments on my manuscript. The conclusion was improved and more detailed by the results and hypothesis in the revised manuscript from line 398 to line 412.
(5) Please, describe in more details the role of ATP for calcium conduction, calcium signals and MAPK cascade signals.
Reply: We sincerely appreciate the valuable comments. The more details of the role of ATP for calcium conduction, calcium signals and MAPK cascade signals were redescribed in the revised manuscript in line 248-256.
(6) P. 11. Lines 226-228. Why did authors describe role of ATP synthase in tumor?
Reply: We appreciate the reviewer’s insightful suggestion. we describe the role of ATP synthase in tumor cells to expand the diversity of ATP functions and provide references for studies in plants such as ‘tumor cells contain ATP synthase, but because the content on the cell surface was not the same’.
(7) P. 11. Lines 229-231. Why did authors describe role of ATP synthase in lipid metabolism and development of anti-obesity drugs?
Reply: We thank the reviewer for pointing this out. Natural rubber biosynthesis occurred on rubber particles that only had a single layer of cytomembrane whose main components are lipids. Based on ‘the role of ATP synthase in lipid metabolism of anti-obesity drugs’, we speculated that ATP synthase is located in the membrane to provide energy for the laticifer to synthesize natural rubber.
To sum up, we have made corrected modifications to the revised manuscript. Please do not hesitate to contact us if there are any questions. Thanks again to the reviewer and editor for your hard work! Best wishes to you!
Authors: Bingbing Guo, Songle Fan, Mingyang Liu, Hong Yang, Longjun Dai, Lifeng Wang

Round 2
Reviewer 3 Report
Comments and Suggestions for Authors
The authors considered my questions. I have no other comments.
Author Response
Dear Reviewer:
Thanks very much for your kind work of our paper. On behalf of my co-authors, we would like to express our great appreciation to you.
Thank you and best regards.
Yours sincerely,
Authors: Bingbing Guo, Songle Fan, Mingyang Liu, Hong Yang, Longjun Dai, Lifeng Wang.